# *Mamestra brassicae* Multiple Nucleopolyhedroviruses Prevents Pupation of *Helicoverpa armigera* by Regulating Juvenile Hormone Titer

**DOI:** 10.3390/insects15030202

**Published:** 2024-03-18

**Authors:** Yanqing Yang, Jinping Dai, Guozhi Zhang, Deepali Singh, Xiaoxia Zhang, Zhenpu Liang

**Affiliations:** 1Institute of Microbial Application, Xinjiang Academy of Agricultural Sciences, Urumqi 830000, China; 2College of Life Sciences, Henan Agricultural University, Zhengzhou 450002, China; 3School of Biotechnology, Gautam, Buddha University, Greater Noida 201312, India; deepali@gbu.ac.in

**Keywords:** MbMNPV, *Helicoverpa armigera*, RNA-seq, juvenile hormone (JH), JHE

## Abstract

**Simple Summary:**

Baculovirus infection can prevent the pupation of insects. However, the molecular mechanism of baculovirus preventing the pupation of larvae by regulating the Juvenile hormone (JH) pathway is still unclear. In this study, we examined the interactive relationship of *Mamestra brassicae* multiple nucleopolyhedroviruses (MbMNPV) and *Helicoverpa armigera* (*H. armigera*). We found that MbMNPV infection caused an increased JH titer and inhibited the expression levels of *juvenile hormone esterase* (*JHE*) and *juvenile hormone epoxide hydrolase* (*JHEH*). Our studies further proved that the JH is mainly degraded by JHE in *H. armigera* larvae. Knocking down of *HaJHE* promoted MbMNPV replication. These findings suggest that the infection of *H. armigera* larvae by MbMNPV leads to an increased JH titer by inhibiting the expression of *JHE*, which prevents pupation of *H. armigera* and promotes MbMNPV replication.

**Abstract:**

Baculovirus infection can prevent the pupation of insects. Juvenile hormone (JH) plays a vital role in regulating insect molting and metamorphosis. However, the molecular mechanism of baculovirus preventing the pupation of larvae by regulating the Juvenile hormone (JH) pathway is still unclear. In this study, we found that the *Mamestra brassicae* multiple nucleopolyhedroviruses (MbMNPV) infection prolonged the larval stage of fourth instar *Helicoverpa armigera* (*H. armigera*) by 0.52 d and caused an increase in JH titer. To identify the genes that contribute to the JH increase in *H. armigera*-MbMNPV interaction, we analyzed mRNA expression profiles of the fat bodies of *H. armigera* infected by MbMNPV. A total of 3637 differentially expressed mRNAs (DE-mRNAs) were filtered out through RNA-seq analysis. These DE-mRNAs were mainly enriched in Spliceosome, Ribosome biogenesis in eukaryotes, Aminoacyl-tRNA biosynthesis, Mismatch repair, and RNA degradation signaling pathway, which are related to the virus infection. Real-time PCR was used to verify the RNA sequencing results. To find out which genes caused the increase in JH titer, we analyzed all the DE-mRNAs in the transcriptome and found that the *JHE* and *JHEH* genes, which were related to JH degradation pathway, were down-regulated. *JHE* and *JHEH* genes in the larvae of MbMNPV-infected group were significantly down-regulated compared with the control group by RT-qPCR. We further proved that the JH is degraded by JHE in *H. armigera* larvae by RNAi, ELISA, RT-qPCR and bioassay, while the hydrolysis of JH by JHEH in *H. armigera* larvae can almost be ignored. Knocking down of *HaJHE* promoted the expression of the JH receptor gene *Met* and the downstream gene *Kr-h1*, and the replication of MbMNPV. This study clarified that JH is mainly degraded by JHE in *H. armigera* larvae. The MbMNPV infection of *H. armigera* larvae leads to the increase of JH titer by inhibiting the expression of *JHE*. The increase in JH titer promotes the expression of the JH receptor gene *Met* and the downstream gene *Kr-h1*, which prevents the pupation of *H. armigera*, and promotes MbMNPV replication. This study provides new insights into *H. armigera* and MbMNPV interaction mechanisms.

## 1. Introduction

*Helicoverpa armigera* (*H. armigera*) is an insect pest [1] that harms several crops, such as wheat, maize, tomato, pepper, and cotton, leading to heavy crop losses worldwide [2]. Control of the pest mainly relies on chemical pesticides, which has led to increasingly severe issues, such as food safety, ecological environment, and pest resistance. Hence, there is an urgent requirement to develop new, efficient, and safe bio-pesticides.

Baculoviruses have high host specificity and are deemed promising novel viral insecticides [3]. So far, only three kinds of broad-spectrum baculoviruses have been identified, including *Mamestra brassicae* multiple nucleopolyhedrovirus, *Autographa californica* multiple nucleopolyhedrovirus, and the *Anagrapha falcifera* multinucleocapsid [4,5,6,7]. *Mamestra brassicae* multiple nucleopolyhedrovirus (MbMNPV), a member of Baculoviridae, was isolated from the cabbage armyworm *Mamestra brassicae* belonging to the Noctuidae family, which is characterized by a wide host spectrum because it can infect 32 species of Lepidoptera pests [4]. MbMNPV has been developed as a commercial bio-pesticide in China [4].

Insect viruses, as the most important pathogens to Lepidoptera insects, can alter the development and physiological status of their hosts [8,9,10]. Insects infected with exogenous viruses typically exhibit developmental defects, such as prolonged larval stage and pupation failure [11,12]. Studies have found that the *egt* gene encoded by baculovirus can regulate the activity of ecdysone, and prevent infected larvae from molting or pupation [12,13]. Juvenile hormone (JH) can regulate the insect growth, development, reproduction, and metamorphosis [14]. Some of the genes in the JH signaling pathway have been reported, such as *juvenile hormone esterase* (*JHE*) [15], juvenile hormone epoxide hydrolase (*JHEH*) [16], juvenile hormone receptor *Methoprene-tolerant* (*Met*) [17], and *Krüppel-homolog 1* (*Kr-h1*) [18]. However, the molecular mechanism of how the baculovirus prevents the pupation of larvae by juvenile hormone (JH) pathway is still unclear.

JHE plays a crucial role in controlling the JH titer in insects, especially during the late larval stage. Insect metamorphosis requires a low JH concentration [19]. The enzyme activity, distribution, and content of JHE vary depending on the type, developmental stage, and tissue organ of insects. JHEH is responsible for the irreversible hydrolysis of JH epoxides, and the metabolite JHA has been shown to have hormonal activity [20,21]. The functional study of the role and regulation of JHE and JHEH in *H. armigera* has not been reported yet. In insects, molting and metamorphosis can be regulated through the JH-Met-Kr-h1 signaling pathway. *Kr-h1*, an early response gene induced by JH, is also involved in insect metamorphosis and reproduction. JH can induce the expression of downstream gene *Kr-h1* through the JH receptor Met, and thereby it plays a vital role in regulating insect metamorphosis [22,23]. The fat body of insects is a vital organ for biosynthesis, metabolism, detoxification, and control of insect ecdysis. It has become a significant research model for the study of metabolic disorders [24].

This study focuses on understanding the mechanism of how the baculovirus prevents the pupation of larvae by controlling the JH pathway. Our results provide a new insight into the role of JHE in *H. armigera*-MbMNPV interactions, which is helpful in understanding the host–baculovirus relation and can be a candidate to develop newer insecticides.

## 2. Materials and Methods

### 2.1. Insects and Viruses

The eggs of *H. armigera* were purchased from Jiyuan Baiyun Industry Co., Ltd. (Jiyuan, China). Insects were fed with an artificial diet and reared in the laboratory at 27 ± 2 °C, 60 ± 2% relative humidity, with a photoperiod of 16:8 (light:dark) [25].

MbMNPV culture was available in our laboratory and was propagated by feeding to third instar *H. armigera* larvae. Occlusion bodies (OBs) of MbMNPV were purified as described by Kyei-Poku & Kunimi [26]. The OBs were purified from MbMNPV-infected cadavers by differential centrifugation followed by sucrose density gradient ultracentrifugation.

### 2.2. Virus Infection and Juvenile Hormone Determination

The newly molted third instar larvae were fed with an artificial diet contaminated with MbMNPV occlusion bodies (30 OBs) after starving for 5 h. Upon consumption of the diets, the larvae were fed with fresh diets without exposure to any pathogens. Mock-infected larvae were fed with the same amounts of diet, but coated with distilled water. Thirty larvae per group were evaluated for larval weight, pupation rate, emergence rate, and larval mortality. The experiments were performed with three independent biological replicates.

In addition, the newly molted fourth instar larvae were fed with artificial diets contaminated with MbMNPV occlusion bodies (OBs, 1 × 10^8^ OBs/larva) after starving for 20 h. Thirty larvae per group were evaluated at fourth instar larval stage. JH titer in fat body and hemolymph at 24 h, 48 h, 72 h, and 96 h post-infection (hpi) was determined by ELISA using the JH ELISA kits from Jiangsu Meimian Industrial Co., Ltd. (Yancheng, Jiangsu, China) [27,28] according to the manufacturer’s instructions. The experiments were performed with three independent biological replicates and technical replicates.

### 2.3. RNA Extraction and Quality Control

The fat body of MbMNPV-infected and mock-infected larvae consisted of three repeats labeled as MbMNPV_1, MbMNPV_2, and MbMNPV_3, and CK_1, CK_2, and CK_3, respectively. At 72 h post-infection (hpi), total RNA was extracted using the TRIZOL reagent (Invitrogen, Carlsbad, CA, USA) according to the manufacturer’s protocol. The RNA quality and purity were evaluated using agarose gel electrophoresis, a Nanodrop microvolume spectrophotometer and an Agilent 2100 Bioanalyzer (Agilent Technologies, Santa Clara, CA, USA).

### 2.4. Library Preparation, mRNAs Sequencing, and Analysis

The libraries were sequenced on the Illumina Novaseq 6000 platform and 150 bp paired-end reads were generated. Raw reads were processed using fastp software (v 0.20.1) [29] and the low quality reads were removed to obtain the clean reads. The clean reads were mapped to the reference genome using HISAT2 software (v 2.1.0) [30] and the expression level of each gene was normalized by the FPKM (transcript per million) method [31]. Differential expression analysis was performed using the DESeq2 software (v 1.22.2) [32]. The differentially expressed mRNAs (DE-mRNAs) were selected with thresholds of *q*-value < 0.05 and |log2 (foldchange)| > 1. GO enrichment [33] and KEGG pathway enrichment analysis [34] of DE-mRNAs were respectively performed using R (v 3.2.0) based on the hypergeometric distribution. Top 30 GO terms and Top 20 of KEGG enrichment were considered to be enriched.

### 2.5. Validation of mRNA by Real-Time Quantitative PCR

To validate the RNA-seq data, three mRNAs were randomly selected for RT-qPCR confirmation with ChamQ™ Universal SYBR^®^ RT-qPCR master mix (Vazyme, Nanjing, China). Total RNAs were extracted using RNA isolator Total RNA Extraction Reagent (Vazyme, China). For gene quantification, the cDNA was prepared using HiScript^®^ III All-in-one RT SuperMix Perfect for RT-qPCR (Vazyme, Nanjing, China) with the gDNA wiper. The PCR was performed with a StepOnePlus™ Real-Time PCR Systems (Applied Biosystems, Thermo Fisher Scientific, Waltham, MA, USA) under the following conditions: 95 °C for 30 s, followed by 40 cycles of 95 °C for 10 s and 60 °C for 30 s. The specificity of each primer pair was ensured by analyzing the melting curve. The experiments were performed with three independent biological replicates and technical replicates. The relative quantity of mRNAs was analyzed by RT-qPCR, and β-actin was used as the reference gene. All the primers were synthesized by the Sangon Biotech Co., Ltd. (Shanghai, China) (Appendix A). The data were analyzed by the 2^−ΔΔCT^ method.

### 2.6. The Expression Patterns, Synthesis of dsRNA, RNA Interference

The expression patterns of *JHE* and *JHEH* in the fat bodies and midgut at different developmental stages of *H. armigera* larvae, viz. the last larval instar (fifth instar) molting stage (5L-M), the first day after last larval instar molting stage (5L-1), the second day after last larval instar molting stage (5L-2), the third day after last larval instar molting stage (5L-3), the fourth day after last larval instar molting stage (5L-4), and the fifth day after last larval instar molting stage (5L-5), were determined by RT-qPCR.

The synthesis and microinjection of dsRNA were performed as per the standard protocol (Ref). Briefly, target fragments of *GFP* (green fluorescent protein), *HaJHE*, and *HaJHEH* for dsRNA synthesis were amplified using specific primers. The PCR products were sub-cloned into the 5 min TA/Blunt-Zero Cloning Kit (Vazyme, China) and used as templates for target sequence amplification. dsRNAs were synthesized in vitro using T7 RNAi Transcription Kit (Vazyme, China), with specific primers fusing the T7 promoter at the 5′ end. The purity and yield of the dsRNA was checked on 1.0% agarose gel and spectrophotometer, respectively. The primers used for the synthesis of dsRNAs are shown in Appendix A. In the knockdown experiment, newly molted last larval instar larvae were incubated for 30 min on ice, followed by injection of 10 μg single dsRNA solution (ds *GFP*, ds *HaJHE*, and ds *HaJHEH*) using a 10 µL microsyringe (Hamilton) in their last leg.

The expression level of *JHE*, *JHEH*, *Met*, and *Kr-h1* gene, and JH titer in larvae injected with dsRNA (ds *GFP*, ds *HaJHE*, and ds *HaJHEH*) was detected by RT-qPCR and ELISA, respectively.

To study the effects of knocking down of *HaJHE* and *HaJHEH* in *H. armigera* larvae, the larval weight, feed intake, pupation rate, and pupation time of last larval instar *H. armigera* larvae injected with ds *GFP*, ds *HaJHE*, and ds *HaJHEH* respectively were measured by bioassay.

### 2.7. Viral Infection

We further investigated the impact of *JHE* knocking down on MbMNPV replication. Larvae on the first day after the last larval instar molting stage injected with dsRNA solution were fed with fresh artificial diets for 4 h. After overnight starvation, the larvae were fed with artificial diets coated with a MbMNPV suspension (OBs; 1 × 10^7^ OBs/larva) followed with feeding of fresh diets without exposure to any pathogens. Subsequently, three larvae infected with mock and MbMNPV were randomly selected at 48 h, and the tissues of the fat body were dissected. To reveal the impact of knocking down of *HaJHE* on MbMNPV infection, the relative expression of MbMNPV *polyhedrin* gene was performed by RT-qPCR for the mock-infected and MbMNPV-infected group.

### 2.8. Data Analysis

All statistical analyses in this study were performed with Prism software and the data were represented as means ± SEM. The statistical significance was determined by Student’s *t*-test for unpaired comparisons between two different groups, and *p* < 0.05 was regarded as statistically significant.

## 3. Results

### 3.1. MbMNPV Infection Prolonged the Larval Stage and Caused the Increase of JH Titer

The larval weight, pupation rate, emergence rate, larval mortality, length of fourth instar larval stage, and JH titer of the mock-infected and MbMNPV-infected larvae were evaluated. Compared with the mock-infected group, the larval weight, pupation rate, and emergence rate of the MbMNPV-infected larvae had a significant reduction (Figure 1A–C). There was a significant increase in the larval mortality, length of the fourth instar larval stage, and JH titer of the MbMNPV-infected larvae (Figure 1D–F). The fourth instar larval stage of the MbMNPV-infected larvae was extended by 0.52 d (Figure 1E). This indicated that MbMNPV infection inhibited the growth of *H. armigera*, prolonged the larval stage, and caused the increase of JH titer.

### 3.2. RNA-Seq Sequencing and Data Analysis

Six mRNA libraries (MbMNPV-infected and mock-infected larvae, three repeats labeled as MbMNPV_1, MbMNPV_2, and MbMNPV_3, and CK_1, CK_2, and CK_3, respectively) were sequenced and the quality of the sequencing data was analyzed (Table 1). Clean Reads were distributed at 47.24 M–49.58 M, with Q30 of 87.79–90.36%.

The DE-mRNAs between the MbMNPV-infected group and the control were screened based on FPKM values (*q*-value < 0.05 and |log2 (foldchange)| > 1). A total of 3637 mRNAs were differentially expressed, including 1634 up-regulated and 2003 down-regulated mRNAs (Figure 2A). The volcanic and heat maps (Figure 2C) of the normalized expression of mRNAs were generated and the expression patterns of the DE-mRNAs expressed during MbMNPV infection were analyzed.

GO and KEGG analyses were performed to investigate the function of DE-mRNAs (Figure 2B,D). GO analysis indicated that the DE-mRNAs were mainly enriched in the following terms: mitochondrial translation (GO: 0032543), mRNA splicing via Spliceosome (GO: 0000398), and rRNA processing (GO: 0006364), under biological process category; nucleolus (GO: 0005730), mitochondrial large ribosomal subunit (GO: 0005762), and mitochondrion (GO: 0005739), under cellular component category; RNA binding (GO: 0003723), unfolded protein binding (GO: 0051082), and ATP binding (GO: 0005524), under molecular function category (Figure 2B).

In the KEGG pathway analysis, Aminoacyl-tRNA biosynthesis (ID: haw00970), RNA polymerase (ID: haw03020), Ribosome biogenesis in eukaryotes (ID: haw03008), Spliceosome (ID: haw03040), Mismatch repair (ID: haw03430), and RNA degradation (ID: haw03018) were significantly enriched (Figure 2D).

### 3.3. MbMNPV Infection Inhibited the Expression of JHE and JHEH

Three mRNAs (*ACADSB, bhmt* and *selenbp1-a*) were randomly selected from the DE-mRNAs to verify the sequencing data by RT-qPCR. The results showed that the expression of these three mRNAs in the MbMNPV-infected group was significantly down-regulated compared with the control group (Figure 3). The RT-qPCR results were consistent with that of the sequencing data, indicating that the sequencing data were reliable.

To identify the genes that caused the increase of JH titer, we analyzed all 3637 DE-mRNAs in the transcriptome and found that the *JHE* and *JHEH* genes, which were related to JH degradation pathway, were down-regulated. By RT-qPCR, it was confirmed that the *JHE* and *JHEH* genes in the larvae of MbMNPV-infected group were significantly down-regulated compared with the control (Figure 4). The RT-qPCR results were consistent with that of the sequencing results, indicating that MbMNPV infection might upregulate JH titer by inhibiting the expression of *JHE* and *JHEH*.

### 3.4. Effects of Knocking down of HaJHE and HaJHEH on the Pupation of H. armigera

To study the effects of *HaJHE* and *HaJHEH* knocking down in *H. armigera* larvae, ds*HaJHE*, and ds*HaJHEH* were injected into the *H. armigera*, respectively. After successful knockdown of *HaJHE* and *HaJHEH* (Figure 5A,B), we observed a significant increase of JH level at 48 h after injection of ds*HaJHE*, but there was no significant difference in the case of ds*HaJHEH* (Figure 5C). The relative expression levels of *Met* and *Kr-h1* in *H. armigera* injected with ds*HaJHE* were significantly increased (Figure 5D,E), while those of *H. armigera* injected with ds*HaJHEH* were not different from the control group (Figure 5D,E). In addition, we measured the larval weight, feed intake, pupation rate, and pupation time of last larval instar *H. armigera* larvae injected with ds*GFP*, ds*HaJHE*, and ds *HaJHEH*, respectively. The results showed that in comparison with the control group, the larval weight and feed intake of larvae injected with ds*HaJHE* and ds*HaJHEH* decreased respectively (Figure 5F,G). The bioassay results showed that in comparison with the control group, the larvae injected with ds*HaJHE* had a delayed pupation of 0.43 d (Figure 5H,J,K), while the larvae injected with ds*HaJHEH* had a normal pupation without any delay (Figure 5I–K). In conclusion, these results show that the knockdown of *HaJHE* delayed the pupation of *H. armigera*, indicating that *H. armigera* larvae mainly degrade JH by JHE.

### 3.5. Detection of Expression Patterns of JHE and JHEH

In order to explore the expression patterns of *JHE* and *JHEH* in *H. armigera*, the temporal and spatial expression patterns of *JHE* and *JHEH* in the last larval instar *H. armigera* larvae were analyzed by RT-qPCR (Figure 6). The time expression pattern results showed that *JHE* and *JHEH* reached their peak expression levels at the second day and the first day after last larval instar molting stage, respectively.

The spatial expression pattern results showed that at the 5L-1, 5L-2, 5L-3, and 5L-5 stages, the expression levels of *JHE* and *JHEH* in fat bodies were higher than those in the midgut. The above results indicate that *JHE* and *JHEH* have a spatiotemporal expression specificity in *H. armigera*.

### 3.6. Knockdown of HaJHE Promoted the MbMNPV Infection

Since MbMNPV infection can inhibit the expression of *HaJHE* and prevent the pupation of *H. armigera*, we further investigated the impact of *HaJHE* knock down on MbMNPV replication. The relative expression of the MbMNPV *polyhedrin* gene was performed by RT-qPCR for the mock-infected and MbMNPV-infected group. The results showed that in comparison with the control, the expression level of the MbMNPV *polyhedrin* gene in the *HaJHE*-knocked down larvae increased by 61% (Figure 7), indicating that the knocking down of *HaJHE* promoted the MbMNPV infection.

## 4. Discussion

Baculoviruses, which are the effective biological insecticides for controlling various Lepidoptera pests, are characterized by vertical transmission and sustained control effects. In this study, the bioassays showed that MbMNPV infection had a significant inhibiting effect on *H. armigera* development, providing a theoretical basis for the application of MbMNPV (Figure 1A–D). The metabolism and synthesis of JH play a pivotal role in regulating insect molting and metamorphosis [35]. Studies have shown that insects infected with exogenous viruses typically exhibit developmental defects and prolonged larval stage [11,12]. It has been reported that *Choristoneura fumiferana* entomopoxvirus (CfEPV) infection can induce the change of JH level, and prolongs the larval stage in larval *Mythimna separata* [36], spruce budworm, and *Choristoneura fumiferana* [37], which is consistent with the results of the current study: MbMNPV infection prolonged the larval stage and induced the increase of JH titer in *H. armigera* (Figure 1E–G).

To identify the genes that contribute to the increase of JH titer in MbMNPV-infected *H. armigera*, we performed RNA sequencing of the fat bodies of MbMNPV-infected *H. armigera*. A total of 3637 DE-mRNAs were obtained, of which 1634 were up-regulated and 2003 were down-regulated. Some DE-mRNAs are enriched in pathways related to viral infection, such as Spliceosome [38], Ribosome biogenesis in eukaryotes [39], Aminoacyl-tRNA biosynthesis [40], and RNA degradation [41] (Figure 2). Research has shown that the dengue virus NS5 protein intrudes in the cellular Spliceosome and modulates splicing [42]. Hepatitis B virus X protein and c-Myc cooperate with promoting ribosome biogenesis [43]. HIV-1 utilizes dynamic aminoacyl-tRNA synthetase complexes to promote virus replication [44]. The RNA degradation pathway participates in PPARα-regulated anti-oral tumorigenesis [45]. The degradation and diffusion of viral RNA are bottlenecks in the efficiency of influenza A virus infection [46]. Transcriptome results indicate that the MbMNPV infection alters the host gene expression profile and triggers *H. armigera*-MbMNPV interactions.

In this study, *bhmt*, *ACADSB*, and *selenbp1-a* were identified as DE-mRNAs in MbMNPV-infected *H. armigera* (Figure 3), and the GO and KEGG pathways related to these three genes were further analyzed. *bhmt* is enriched into the zinc ion binding pathway (GO: 0008270) and the cysteine and methionine metabolism pathway (KEGG: haw00270). It has been reported that zinc ions can inhibit the main protease of SARS-CoV-2 in vitro and inhibit virus replication [47]. The increased concentration of homocysteine promotes HIV replication [48]. *ACADSB* is enriched in the branched chain amino acid catabolic process (GO: 0009083) and fatty acid degradation pathway (KEGG: haw00071). Branch chain amino acids reduce liver iron accumulation and oxidative stress in mice with hepatitis C virus multi protein expression [49]. Oncolytic avian reovirus σA regulates fatty acid metabolism to increase energy production for virus replication [50]. *selenbp1-a* is enriched in the nucleus (GO: 0005634) and sulfur metabolism pathway (KEGG: haw00920). In general, the nucleus is an important place for virus replication. The disorder of sulfur metabolism in AIDS patients is related to the depletion of cysteine and glutathione in white blood cells [51,52].

In this study, we also screened genes related to JH synthesis and metabolism pathways from 3637 DE-mRNAs and found that MbMNPV infection inhibited the relative expression levels of *JHE* and *JHEH* (Figure 4). JHE has been cloned or characterized in over 17 insect species, including *Bombyx mori* [53] and *Trichoplusia ni* [54] (Lepidoptera), *Aedes aegypti* [55] and *Drosophila* [56] (Diptera), *Tenebrio molitor* [53] and *Leptinotarsa decemlineata* (Say) [57] (Coleoptera). It has been confirmed that depletion of juvenile hormone esterase extends larval growth in *Bombyx mori* [58], which is consistent with the function of JHE in *H. armigera* in this study (Figure 5H,J,K). There are fewer studies on JHEH in comparison to JHE.

The hydrolysis function of JHEH to JH is different in different species. The effect of hydrolysis of JHEH on JH has been confirmed in tobacco hornworm, while its hydrolysis function in honeybee can almost be ignored [59]. This is consistent with our results that the larvae injected with ds*HaJHEH* had a normal pupation without any delay (Figure 5I–K), indicating the hydrolysis function of JH by JHEH in *H. armigera* larvae can be almost ignored and hence the JH in *H. armigera is* mainly degraded by JHE (Figure 5). Interestingly, we found that MbMNPV infection inhibited the weight gain of *H. armigera* (Figure 1A) and the relative expression levels of *JHE* and *JHEH* (Figure 4), while knocking down of *HaJHE* and *HaJHEH* also reduced the feed intake and weight of larvae (Figure 5F,G), indicating that MbMNPV infection inhibited the feed intake and weight gain of larvae by inhibiting the expression of *HaJHE* and *HaJHEH*. Knocking down of *HaJHE* promoted MbMNPV replication (Figure 7), indicating that deficiency of *HaJHE* reduced JH metabolism in *H. armigera* larvae, resulting in the host maintaining a longer larval state and ultimately promoting virus replication.

## 5. Conclusions

In conclusion, we profiled the differentially expressed mRNAs in the fat bodies of MbMNPV-infected *H. armigera* and clarified that JH is mainly degraded by JHE in *H. armigera* larvae. The MbMNPV infection of *H. armigera* larvae leads to the increase of JH titer by inhibiting the expression of *JHE*. The increase of JH titer promotes the expression of the JH receptor gene *Met* and the downstream gene *Kr-h1*, which prevents the pupation of *H. armigera*, and promotes MbMNPV replication. In the future, we will analyze the molecular mechanism of how the MbMNPV infection inhibits JHE expression. This research provides a new insight into the role of JHE in *H. armigera* and MbMNPV interactions, which is helpful in understanding the host–baculovirus correlation from the perspective baculovirus suppressing the insect development.

## Figures and Tables

**Figure 1 insects-15-00202-f001:**
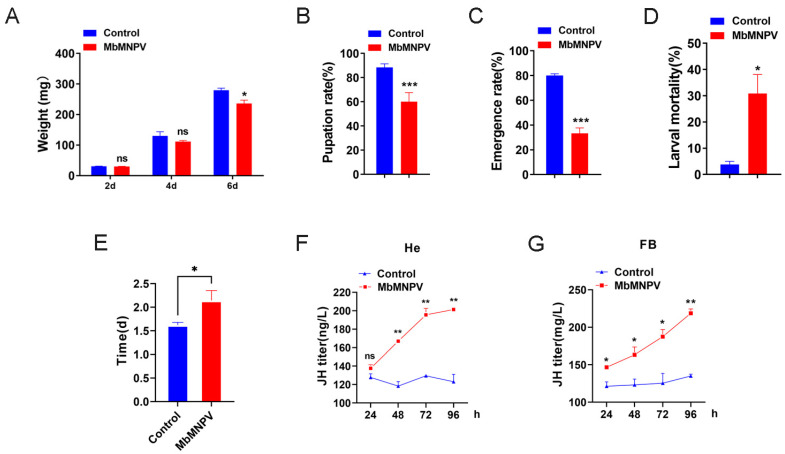
The impact of MbMNPV infection in *H. armigera* larvae. The larval weight (**A**), pupation rate (**B**), emergence rate (**C**), larval mortality (**D**), fourth instar larval stage (**E**), and JH titer in hemolymph (He) (**F**) and fat body (FB) (**G**) in the mock-infected and MbMNPV-infected *H. armigera* larvae were evaluated. ns, *p* > 0.05; ***, *p* < 0.001; **, *p* < 0.01; *, *p* < 0.05; *p* < 0.05 indicated the significant difference (Student’s *t*-test). Each experiment was performed in three replicates, and the data are shown as mean ± s.e.m.

**Figure 2 insects-15-00202-f002:**
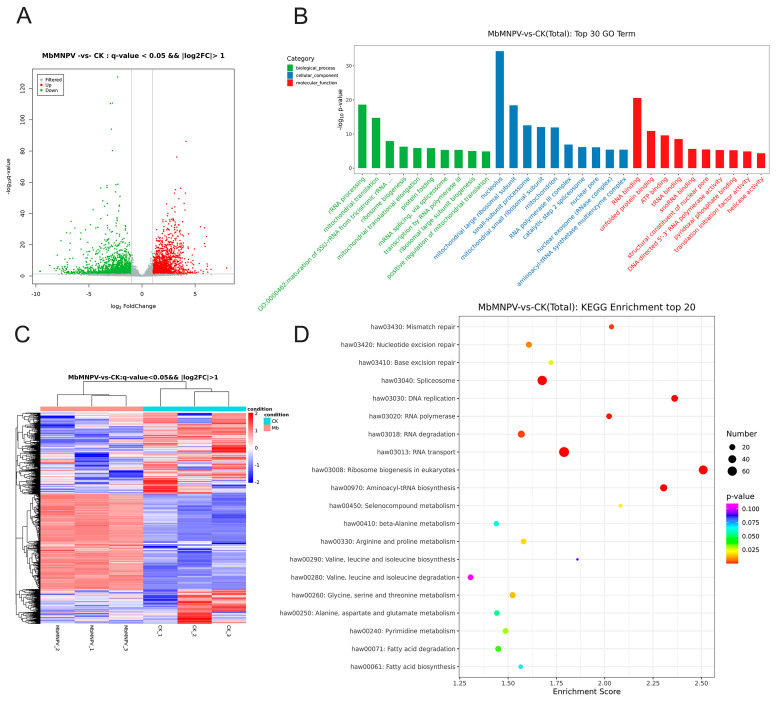
Screening and enrichment analysis of DE-mRNAs in the MbMNPV-infected group compared with control. (**A**) The volcano map of DE-mRNAs. (**B**) Top 30 significantly enriched GO terms of DE-mRNAs in molecular function, cellular component and biological process. (**C**) Hierarchical clustering analysis (heatmap) for DE-mRNAs. (**D**) Top 20 significantly enriched KEGG analyses of DE-mRNAs.

**Figure 3 insects-15-00202-f003:**
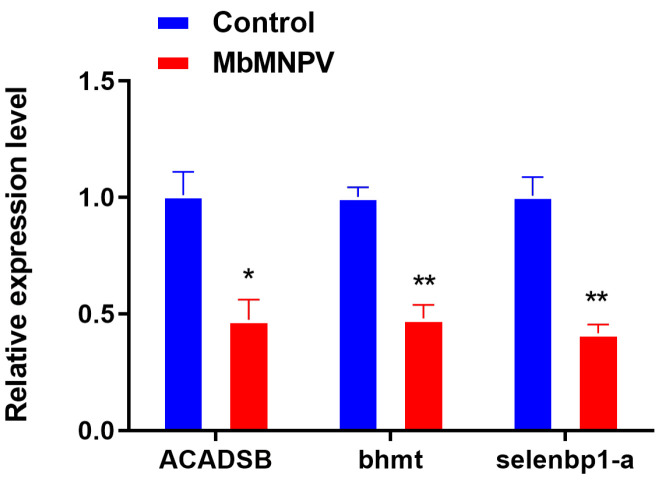
Relative expression of mRNAs in control and MbMNPV. *, *p* < 0.05; **, *p* < 0.01. *p* < 0.05 indicated the significant difference (Student’s *t*-test). Each experiment was performed in three replicates, and data are shown as mean ± s.e.m.

**Figure 4 insects-15-00202-f004:**
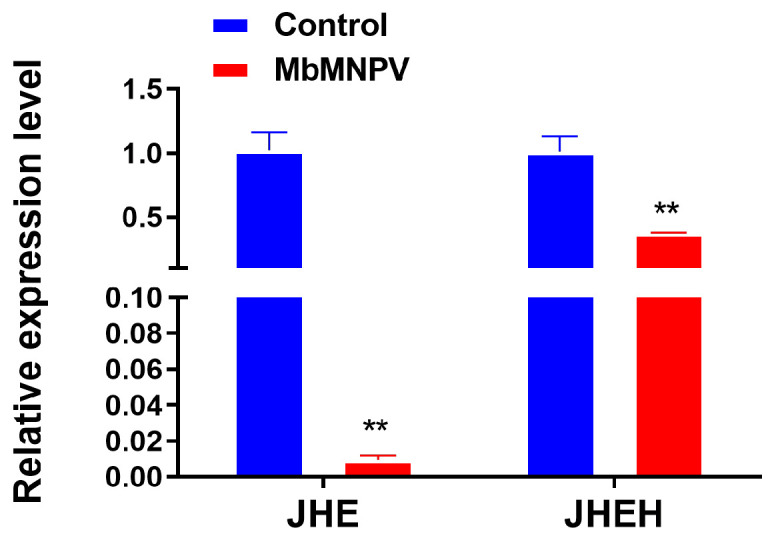
Relative expression of *JHE* and *JHEH* in control and MbMNPV. **, *p* < 0.01. *p* < 0.05 indicated the significant difference (Student’s *t*-test). Each experiment was performed in three replicates, and data are shown as mean ± s.e.m.

**Figure 5 insects-15-00202-f005:**
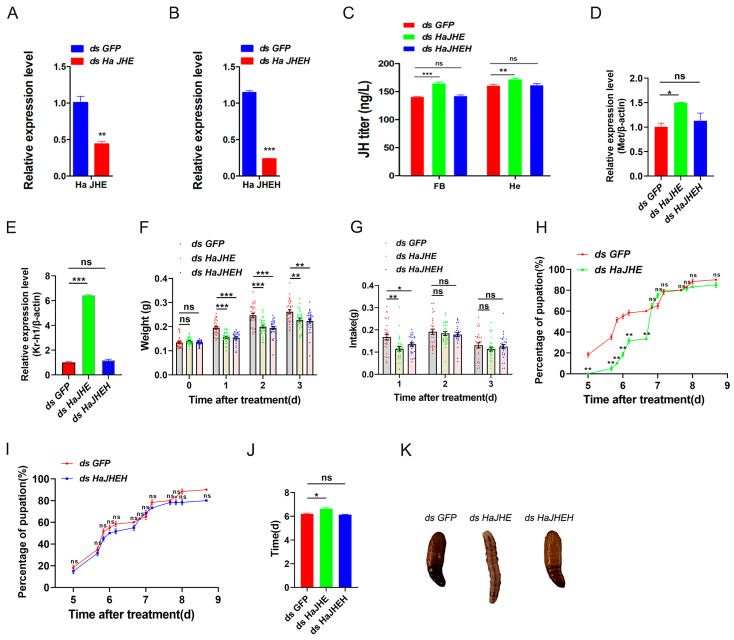
Effects of knocking down of *HaJHE* and *HaJHEH* on the pupation of *H. armigera*. Larvae on the first day after the last larval instar molting stage were injected with dsRNA of *HaJHE* (ds *HaJHE*), *HaJHEH* (ds *HaJHEH*), and dsRNA of Green fluorescent protein (ds *GFP*). (**A**) The expression levels of the *JHE* gene. (**B**) The expression levels of the *JHEH* gene. (**C**) The JH titer. (**D**) The expression levels of the *Met* gene. (**E**) The expression levels of the *Kr-h1* gene. (**F**) The larval weight. (**G**) The feed intake of larvae. (**H**,**I**) The pupation rate. (**J**) The pupation time. (**K**) The larval phenotype after injection of dsRNA. The short vertical line represents a ruler. ns, *p* > 0.05; ***, *p* < 0.001; **, *p* < 0.01; *, *p* < 0.05; *p* < 0.05 indicated the significant difference (Student’s *t*-test). Each experiment was performed in three replicates, and data are shown as mean ± s.e.m.

**Figure 6 insects-15-00202-f006:**
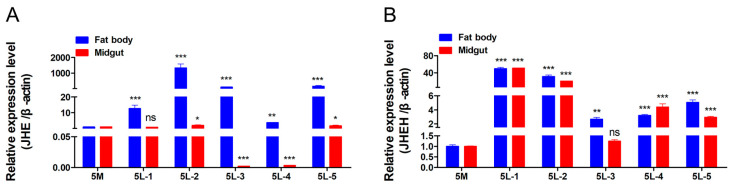
Detection of the expression patterns of *JHE* and *JHEH*. (**A**) The expression patterns of *JHE* gene. (**B**) The expression patterns of *JHEH* gene. ns, *p* > 0.05; ***, *p* < 0.001; **, *p* < 0.01; *, *p* < 0.05; *p* < 0.05 indicated the significant difference (Student’s *t*-test). Each experiment was performed in three replicates, and data are shown as mean ± s.e.m.

**Figure 7 insects-15-00202-f007:**
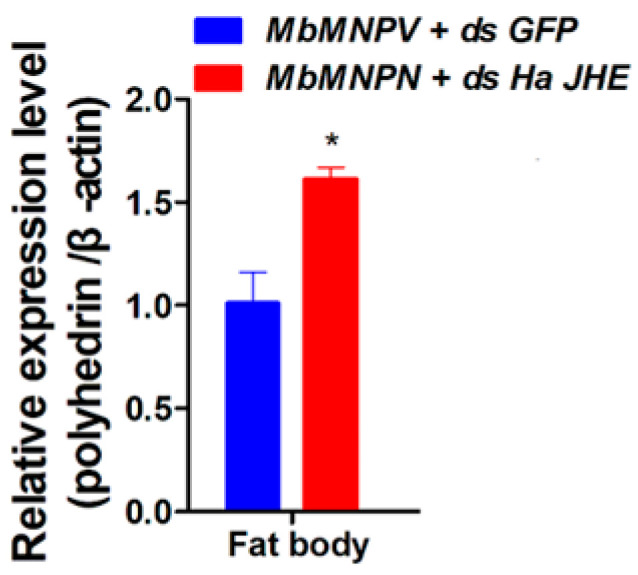
Knockdown of *HaJHE* promoted MbMNPV infection. The relative expression of MbMNPV *polyhedrin* gene was performed by RT-qPCR for the mock-infected and MbMNPV-infected group. *, *p* < 0.05; *p* < 0.05 indicated the significant difference (Student’s *t*-test). Each experiment was performed in three replicates, and data are shown as mean ± s.e.m.

**Table 1 insects-15-00202-t001:** Quality evaluation of sample sequencing data.

Sample	RawReads	RawBases	CleanReads	ValidBases	Q30	GC
CK_1	48.33 M	7.25G	47.27 M	94.87%	87.79%	49.76%
CK_2	48.26 M	7.24G	47.24 M	95.23%	90.05%	49.65%
CK_3	48.54 M	7.28G	47.54 M	95.64%	90.36%	49.31%
MbMNPV_1	50.64 M	7.60G	49.49 M	95.07%	89.39%	49.47%
MbMNPV_2	49.58 M	7.44G	48.39 M	95.04%	90.02%	48.86%
MbMNPV_3	50.75 M	7.61G	49.58 M	94.99%	89.25%	49.82%

## Data Availability

All relevant data are within the article.

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
