# Peer review of "Mamestra brassicae Multiple Nucleopolyhedroviruses Prevents Pupation of Helicoverpa armigera by Regulating Juvenile Hormone Titer"

_insects, 2024, doi:10.3390/insects15030202_

Round 1

Reviewer 1 Report

Comments and Suggestions for Authors

I only have a few comments on this paper, which adds significant new data to the field.  Inhibition of JHE by baculovirus infection of lepidopteran larvae has been well known for many years.  This paper presents a more detailed analysis of gene expression associated with the phenomenon.  RNA sequencing showed that host JHE and JHEH genes were downregulated by MbMNPV infection, allowing JH titres to remain high. Knocking down HaJHE and HaHEH also increased JH levels.

One issue that detracts from the paper is the quality of the figures.  These often appear blurred, particularly Figs 1 and 5.  I printed them on a good quality laser but still had trouble reading some of the information.  On screen they also appear poor.  Increasing their size might help

A further issue concerns Figure 7 and associated text.  This describes increased MbMNPV polyhedrin gene expression but doesn't say how this was measured.  Was this by qPCR of mRNA levels or analysis of proteins?  I suspect it was the former, but this should be clearly stated in the figure legend and the text that describes the figure.

Comments on the Quality of English Language

English quality is good, with only minor edits required as recommended by journal.  

Author Response

Point 1: One issue that detracts from the paper is the quality of the figures. These often appear blurred, particularly Figs 1 and 5. I printed them on a good quality laser but still had trouble reading some of the information. On screen they also appear poor. Increasing their size might help 

Response 1: According to your advice, the sizes of all the images in this article have been improved.

Point 2: A further issue concerns Figure 7 and associated text.  This describes increased MbMNPV polyhedrin gene expression but doesn't say how this was measured.  Was this by qPCR of mRNA levels or analysis of proteins?  I suspect it was the former, but this should be clearly stated in the figure legend and the text that describes the figure.

Response 2: The sentence “The relative expression of MbMNPV polyhedrin gene was performed by RT-qPCR for the mock-infected and MbMNPV-infected group.” has been added in the figure legend and the text that describes this figure.

Reviewer 2 Report

Comments and Suggestions for Authors

     Baculovirus infection affects insect development and physiology. Generally, it delays larval development and inhibits pupation in Lepidptera through disturbances of hormonal status. In the current study, authors confirmed that MbMNPV infection inhibits larval growth and pupation accompanied with increase of juvenile hormone (JH) tire in Helicoverpa armigera. They found that the expressions of genes for two JH-degrading enzymes, JHE and JHEH, were decreased in infected larvae. Knocking down of JHE, but not JHEH, inhibited larval growth and pupation as same as viral infection, suggesting that viral infection controls larval growth by decreasing JHE expression. In addition, authors also showed that JHE is the key enzyme to decrease JH tire at the last larval instar in the normal physiological condition of H. armigera. The study is well designed and organized. This reviewer feels a major concern as follows:

1. Involvement of ecdysteroid signaling

Although authors revealed importance of JH control via JHE expression in the infected larvae, it is also important to see whether ecdysteroid signaling is involved or not. It can be revealed at least partially to show the expression of genes for ecdysteroid signaling such as E75A and ecdysteroid biosynthetic enzymes. 

Other minor concerns are listed below:

1. Antibody for ELISA

Authors performed ELISA to measure JH tire. The reliability of ELISA depends on the quality of antibody. Please state the information of the antibody used for ELISA.

2. Quality of figures

The quality of figures is too low to read. Especially, figure 2 is too small and obscure to provide any information to readers. The labeling of figure 2 is not correct. 

3. The last instar of H. armigera

The fifth instar seems the last larval instar of H. armigera. It should be clearly stated. 

4. Line 59

The cabbage should be cabbage armyworm. 

5. Line 260, 319 

“JH expression” is not correct. It should be JH titer or JH level. 

6. Line 259, 262

Not Figure 6BE but Figure 5BE.

7. Line 356

“Colorado potato beetle” is not academic Latin name. 

Comments on the Quality of English Language

The manuscript is written well as a whole. Several points sounded strange somewhat to this reviewer. They might be better to be fixed. 

Author Response

Point 1: Authors performed ELISA to measure JH tire. The reliability of ELISA depends on the quality of antibody. Please state the information of the antibody used for ELISA.

Response 1: The sentence “JH titer in fat body and hemolymph at 24 h, 48 h, 72 h, and 96 h post-infection (hpi) was determined by ELISA” has been replaced by ” JH titer in fat body and hemolymph at 24 h, 48 h, 72 h, and 96 h post-infection (hpi) was determined by using the JH ELISA kits (MEIMIAN, Jiangsu, China) according to the manufacturer’s instructions”. The information of the antibody used for ELISA is the core information of the company MEIMIAN and it is not publicly disclosed. We provide two references in which this JH ELISA kit was used in the manuscript.

reference:

  1. Yao S, Yang Y, Xue Y, Zhao W, Liu X, Du M, Yin X, Guan R, Wei J, An S. New insights on the effects of spinosad on the development of Helicoverpa armigera. Ecotoxicol Environ Saf. 2021 Sep 15, 221, 112452. doi: 10.1016/j.ecoenv.2021.112452.
  2. Wang SS, Wang LL, Pu YX, Liu JY, Wang MX, Zhu J, Shen ZY, Shen XJ, Tang SM. Exorista sorbillans (Diptera: Tachinidae) parasitism shortens host larvae growth duration by regulating ecdysone and juvenile hormone titers in Bombyx mori (Lepidoptera: Bombycidae). J Insect Sci. 2023 May 1,23(3):6. doi: 10.1093/jisesa/iead034.

Point 2: The quality of figures is too low to read. Especially, figure 2 is too small and obscure to provide any information to readers. The labeling of figure 2 is not correct.

Response 2: According to your advice, the sizes of all the images in this article have been improved. The labeling of figure 2 has been corrected.

Point 3: The fifth instar seems the last larval instar of H. armigera. It should be clearly stated.

Response 3: The phrase “fifth instar” has been replaced by “last larval instar”.

Point 4: The cabbage should be cabbage armyworm.

Response 4: The word “cabbage” has been replaced by “cabbage armyworm”.

Point 5: Line 260, 319 “JH expression” is not correct. It should be JH titer or JH level.

Response 5: The phrase “JH expression” has been replaced by “JH level”.

Point 6: Line 259, 262 Not Figure 6BE but Figure 5BE.

Response 6: The “Figure 6BE” has been replaced by “Figure 5BE”.

Point 7: Line 356“Colorado potato beetle” is not academic Latin name.

Response 7: The phrase “Colorado potato beetle” has been replaced by “Leptinotarsa decemlineata (Say)”.

Point 8: Although authors revealed importance of JH control via JHE expression in the infected larvae, it is also important to see whether ecdysteroid signaling is involved or not. It can be revealed at least partially to show the expression of genes for ecdysteroid signaling such as E75A and ecdysteroid biosynthetic enzymes. 

Response 8: Studies have found that the egt gene encoded by baculovirus can regulate the activity of ecdysone, and prevents infected larvae from molting or pupation. However, the molecular mechanism of baculovirus preventing pupation of larvae by regulating the Juvenile hormone (JH) pathway is still unclear. In this study, we studied the interactive relationship of Mamestra brassicae multiple nucleopolyhedroviruses (MbMNPV) and Helicoverpa armigera (H. armigera) from JH pathway. The research background of this section has been reflected in the introduction section.

Round 2

Reviewer 2 Report

Comments and Suggestions for Authors

     Points that concern this reviewer in the revised manuscript are as follows:

1. References

References are repeated at lines 397-532 and 537-665. The former one seems the old version. 

2. Notation of gene names

Some genes names are italicized, and some are not. For example, Kr-h1 at line 81 are italicized, but Kr-h1 at line 82 are not. They should be in the same way.

Author Response

Point 1: References are repeated at lines 397-532 and 537-665. The former one seems the old version. 

Response 1: The former one is the old version and it has been deleted.

Point 2: Some genes names are italicized, and some are not. For example, Kr-h1 at line 81 are italicized, but Kr-h1 at line 82 are not. They should be in the same way.

Response 2: According to your advice, genes names in this manuscript have been italicized.
